# Stochastic Second-Order Methods Improve Best-Known Sample Complexity of SGD for Gradient-Dominated Functions

**Saeed Masiha**[*]
College of Management of Technology
EPFL, Lausanne, Switzerland
`mohammadsaeed.masiha@epfl.ch`

**Saber Salehkaleybar**[*]
School of Computer and Communication Sciences
EPFL, Lausanne, Switzerland
`saber.salehkaleybar@epfl.ch`

**Niao He**
Department of Computer Science
ETH, Zurich, Switzerland
`niao.he@inf.ethz.ch`

**Negar Kiyavash**
College of Management of Technology
EPFL, Lausanne, Switzerland
`negar.kiyavash@epfl.ch`

**Patrick Thiran**
School of Computer and Communication Sciences
EPFL, Lausanne, Switzerland
`patrick.thiran@epfl.ch`

## Abstract

We study the performance of Stochastic Cubic Regularized Newton (SCRN) on a class of functions satisfying gradient dominance property with $1 \leq \alpha \leq 2$ which holds in a wide range of applications in machine learning and signal processing. This condition ensures that any first-order stationary point is a global optimum. We prove that the total sample complexity of SCRN in achieving $\epsilon$-global optimum is $\mathcal{O}(\epsilon^{-7/(2\alpha)+1})$ for $1 \leq \alpha < 3/2$ and $\tilde{\mathcal{O}}(\epsilon^{-2/(\alpha)})$ for $3/2 \leq \alpha \leq 2$. SCRN improves the best-known sample complexity of stochastic gradient descent. Even under a weak version of gradient dominance property, which is applicable to policy-based reinforcement learning (RL), SCRN achieves the same improvement over stochastic policy gradient methods. Additionally, we show that the average sample complexity of SCRN can be reduced to $\mathcal{O}(\epsilon^{-2})$ for $\alpha = 1$ using a variance reduction method with time-varying batch sizes. Experimental results in various RL settings showcase the remarkable performance of SCRN compared to first-order methods.

## 1 Introduction

Consider the following unconstrained stochastic non-convex optimization problem:

$$\min_{\mathbf{x} \in \mathbb{R}^d} F(\mathbf{x}) := \mathbb{E}_\xi[f(\mathbf{x}, \xi)], \tag{1}$$

where the random variable $\xi$ is sampled from an underlying distribution $P_\xi$. In order to optimize the objective function $F(\mathbf{x})$, we have access to the first and second derivatives of stochastic function $f(\mathbf{x}, \xi)$. The above optimization problem covers a wide range of problems, from the offline setting where the objective function is minimized over a fixed number of samples, to the online setting where the samples are drawn sequentially.

---

[*]equal contribution

36th Conference on Neural Information Processing Systems (NeurIPS 2022).

In the deterministic case (where we have access to the derivatives of $F(\mathbf{x})$), the gradient descent (GD) algorithm in the non-convex setting only guarantees convergence to a first-order stationary point (FOSP) (i.e., a point $\mathbf{x}$ such that $\|\nabla F(\mathbf{x})\| = 0$), which can be a local minimum, a local maximum, or a saddle point. In contrast, second-order methods accessing the Hessian of $F$ (or Hessian of $f(\mathbf{x}, \xi)$ in the stochastic setting) can exploit the curvature information to effectively escape saddle points and converge to a second-order stationary point (SOSP) (i.e., such that $\|\nabla F(\mathbf{x})\| = 0$, $\nabla^2 F(\mathbf{x}) \succeq 0$). In their seminal work, Nesterov and Polyak [25] proposed the so-called cubic-regularized Newton (CRN) algorithm which exploits Hessian information and globally converges to an SOSP at a sub-linear rate of $\mathcal{O}(1/k^{2/3})$, where $k$ is the number of iterations.

In recent years, the performance of stochastic CRN (SCRN) for general non-convex functions has been the focus of several studies (for more details, see the related work in Section 1.1). A variance-reduced version of SCRN [3] can find $(\epsilon, \gamma)$-SOSP (i.e., a point $\mathbf{x}$ such that $\|\nabla F(\mathbf{x})\| \leq \epsilon$ and $\nabla^2 F(\mathbf{x}) \succeq -\gamma I$) with the sample complexity of $\tilde{\mathcal{O}}(\epsilon^{-3})$. Moreover, this rate is optimal for achieving $\epsilon$-approximate FOSP (i.e., a point $\mathbf{x}$ such that $\|\nabla F(\mathbf{x})\| \leq \epsilon$) and it cannot be improved using any stochastic $p$-th order methods for $p \geq 2$ [3].

Nesterov and Polyak [25] studied CRN under the gradient dominance property (See Assumption 2). Under Assumption 2 with $\alpha = 2$, they showed that iterates of $F(\mathbf{x}_t) - \min_{\mathbf{x}} F(\mathbf{x})$ converges to zero super-linearly. The case of $\alpha = 2$ (commonly called Polyak-Łojasiewicz (PL) condition) was originally introduced by Polyak in [28], who showed that GD achieves linear convergence rate. The gradient dominance property or its weak variations are satisfied in a quite wide range of machine learning applications such as neural networks with one hidden layer [18] or ResNet with linear activation [12] (for more details, see Section 1.1). A particularly important application of the weak version of gradient dominance property with $\alpha = 1$ (Assumption 6) is in policy-based reinforcement learning (RL) [41].

Khaled et al. [16] showed that under PL condition, the stochastic GD (SGD) with time-varying step-size returns a point $\hat{\mathbf{x}}$ with a sample complexity of $\mathcal{O}(1/\epsilon)$, to reach $\mathbb{E}[F(\hat{\mathbf{x}})] - \min_{\mathbf{x}} F(\mathbf{x}) \leq \epsilon$. Furthermore, the dependency of the sample complexity of SGD on $\epsilon$ is optimal [26]. Recently, Fontaine et al. [10] obtained a sample complexity of $\mathcal{O}(\epsilon^{-\frac{4}{\alpha}+1})$ for SGD under gradient dominance property with $1 \leq \alpha \leq 2$. This shows that the worst sample complexity occurs for $\alpha = 1$, which is precisely the value of $\alpha$ that finds important applications in policy-based RL. Indeed, under a weak version of gradient dominance property with $\alpha = 1$, it has been shown that stochastic policy gradient (SPG) converges to the optimum point with a sample complexity of $\tilde{\mathcal{O}}(\epsilon^{-3})$ [41, 9].

We know that in the deterministic case, CRN outperforms GD under gradient dominance property for all $\alpha \in [1, 2]$ [25, 50][2]. Therefore, a natural question that arises is whether this holds true in the stochastic setting as well? That is, does SCRN improve upon SGD under the gradient dominance property?

Herein, we address this question. Specifically, our main contributions are as follows:

- We analyze the sample complexity of SCRN under gradient dominance property for $1 \leq \alpha \leq 2$ in order to return an $\epsilon$-global stationary point $\hat{\mathbf{x}}$ satisfying $F(\hat{\mathbf{x}}) - \min_{\mathbf{x}} F(\mathbf{x}) \leq \epsilon$ (in expectation or with high probability). As stated in Table 1, SCRN improves upon the best-known sample complexity of SGD for all $1 \leq \alpha < 2$. The largest improvement is for $\alpha = 1$, and is $\mathcal{O}(\epsilon^{-0.5})$.

- In the setting of policy-based RL, under the weak version of gradient dominance property with $\alpha = 1$ (Assumption 6), we show that SCRN achieves a sample complexity of $\tilde{\mathcal{O}}(\epsilon^{-2.5})$, improving over the best-known sample complexity of SPG by a factor of $\tilde{\mathcal{O}}(\epsilon^{-0.5})$.

- We show that an adaptation of a variance-reduced SCRN [47] with time-varying batch sizes further improves the sample complexity of SCRN. For $\alpha = 1$, the average sample complexity is reduced to $\mathcal{O}(\epsilon^{-2})$.

---

[2]In particular, for $\alpha \in [1, 3/2)$, the number of iterations of CRN is $\mathcal{O}(1/\epsilon^{3/(2\alpha)-1})$, for $\alpha = 3/2$ is $\mathcal{O}(\log(1/\epsilon))$, and for $\alpha \in (3/2, 2]$, the number of iterations is $\mathcal{O}(\log \log(1/\epsilon))$. For $\alpha \in [1, 2)$, the number of iterations of GD is $\mathcal{O}(1/\epsilon^{2/\alpha-1})$ and for $\alpha = 2$, is $\mathcal{O}(\log(1/\epsilon))$ [50].

Table 1: Comparison of sample complexities of SGD on Lipschitz gradient functions and SCRN on Lipschitz Hessian functions to achieve $\epsilon$-global stationary point under gradient dominance property with $\alpha \in [1, 2]$. The last column indicates the improvement of SCRN with respect to SGD.

| $\alpha$ | SGD [10] | SCRN (Ours) | Improvement |
|---|---|---|---|
| $[1, \frac{3}{2})$ | $\mathcal{O}(\epsilon^{-4/\alpha+1})$ | $\mathcal{O}(\epsilon^{-7/(2\alpha)+1})$ | $\mathcal{O}(\epsilon^{-1/(2\alpha)})$ |
| $\frac{3}{2}$ | $\mathcal{O}(\epsilon^{-4/\alpha+1}) = \mathcal{O}(\epsilon^{-5/3})$ | $\tilde{\mathcal{O}}(\epsilon^{-7/(2\alpha)+1}) = \tilde{\mathcal{O}}(\epsilon^{-4/3})$ | $\tilde{\mathcal{O}}(\epsilon^{-1/3})$ |
| $(\frac{3}{2}, 2)$ | $\mathcal{O}(\epsilon^{-4/\alpha+1})$ | $\tilde{\mathcal{O}}(\epsilon^{-2/\alpha})$ w.h.p | $\tilde{\mathcal{O}}(\epsilon^{-2/\alpha+1})$ |
| $2$ | $\mathcal{O}(\epsilon^{-1})$ | $\tilde{\mathcal{O}}(\epsilon^{-2/\alpha}) = \tilde{\mathcal{O}}(\epsilon^{-1})$ w.h.p | – |

## 1.1 Related work

**Gradient dominance property and its applications:** The gradient dominance property with $\alpha = 2$ (commonly called PL condition) was originally introduced by Polyak in [28]. It was shown by Karimi et al. [15] to be weaker than the most recent global optimality conditions that appeared in the literature of machine learning [19, 23, 42]. The gradient dominance property is also satisfied (sometimes locally rather than globally, and also under distributional assumptions) for the population risk in some learning models including neural networks with one hidden layer [18], ResNet with linear activation [12], and generalized linear model and robust regression [11]. Moreover, in policy-based reinforcement learning (RL), a weak version of gradient dominance property with $\alpha = 1$ (see Assumption 6) holds for some classes of policies (such as Gaussian policy and log-linear policy).

**Variants of cubic regularized Newton method:** For non-convex optimization, Nesterov and Polyak [25] proposed the CRN algorithm, which converges to a SOSP with the convergence rate of $\mathcal{O}(1/k^{2/3})$ (where $k$ is the number of iterations) by solving a cubic sub-problem in each iteration. Cartis et al. [5] presented an adaptive framework for cubic regularization method. In [17, 39], sub-sampled versions of gradient and Hessian were used in CRN to overcome the computational burden of Hessian matrix evaluations in high dimensional settings. In the context of stochastic optimization, Tripuraneni et al. [33] proposed a stochastic cubic regularization algorithm that obtains $\epsilon$-SOSP with sample complexity of $\tilde{\mathcal{O}}(\epsilon^{-3.5})$. Arjavani et al. [3] improved the sample complexity to $\tilde{\mathcal{O}}(\epsilon^{-3})$ using variance reduction. In the convex setting, Song et al. [30] presented a proximal CRN and its accelerated version, and proved a sample complexity of $\tilde{\mathcal{O}}(\epsilon^{-2})$ to reach $\epsilon$-global stationary point as long as the approximated Hessian in each iteration satisfies certain properties. In finite-sum non-convex setting, Zhou et al. [48] proposed an adaptive sub-sampled CRN method that requires $\tilde{\mathcal{O}}(N + N^{4/5}\epsilon^{-3/2})$ to find $\epsilon$-SOSP, where $N$ is the total number of samples. Sample complexity was further reduced to $\tilde{\mathcal{O}}(N + N^{2/3}\epsilon^{-3/2})$ using various variance reduction methods [36, 45, 49]. To the best of our knowledge, no previous work on analyzing SCRN for gradient-dominant functions exists.

## 1.2 Notations

We adopt the following notation in the sequel. Calligraphic letters (e.g., $\mathcal{S}$) denote spaces. Upper-case bold letters (e.g., $\mathbf{A}$) denote matrices, and the lower-case bold letters (e.g., $\mathbf{x}$) denote vectors. $\|\cdot\|$ denote the $\ell_2$-norm for vectors and the operator norm for matrices ($\|\mathbf{A}\| := \lambda_{\max}(\mathbf{A}^T\mathbf{A})$ where $\lambda_{\max}(\mathbf{X})$ is the maximum eigenvalue of matrix $\mathbf{X}$), respectively. $\mathbf{A} \succeq \mathbf{B}$ indicates that $\mathbf{A} - \mathbf{B}$ is positive semi-definite. We use the notation $\mathcal{O}$ to hide constants, and the notation $\tilde{\mathcal{O}}$ to hide both constants and logarithmic factors. $X \leq_{1-\delta} Y$ denotes that random variable $X$ is less than or equal to random variable $Y$ with the probability at least $1 - \delta$.

## 2 Setup

Recall the stochastic non-convex optimization problem in (1), where the goal is to minimize the objective function $F(\mathbf{x})$ having access to stochastic gradients $\nabla f(\mathbf{x}, \xi)$ and stochastic Hessian matrices $\nabla^2 f(\mathbf{x}, \xi)$. We make the following assumption about the objective function in (1).

**Assumption 1.** *The Hessian of $F$ is Lipschitz continuous with constant $L_2$, i.e.,*

$$\|\nabla^2 F(\mathbf{x}) - \nabla^2 F(\mathbf{y})\| \leq L_2\|\mathbf{x} - \mathbf{y}\|_2, \quad \forall \mathbf{x}, \mathbf{y} \in \mathbb{R}^d. \tag{2}$$

Consider the empirical estimators $\mathbf{g}_t := \frac{1}{n_1}\sum_{i=1}^{n_1}\nabla f(\mathbf{x}_t, \xi_i)$, and $\mathbf{H}_t := \frac{1}{n_2}\sum_{i=1}^{n_2}\nabla^2 f(\mathbf{x}_t, \xi_i)$ where $n_1$ and $n_2$ are the numbers of samples used for estimating the gradient vector and Hessian matrix, respectively.

---

**Algorithm 1** Stochastic cubic regularized Newton method with stopping criterion

---

**Input:** Batch sizes $n_1, n_2$, initial point $\mathbf{x}_0$, accuracy $\epsilon$, cubic penalty parameter $M$, maximum number of iterations $T$

1: $t \leftarrow 1$
2: $\|\mathbf{\Delta}_0\| = \infty$
3: **while** $\|\mathbf{\Delta}_{t-1}\| \geq \sqrt[2\alpha]{\epsilon}$ or $t \leq T$ **do**
4:     $\mathbf{g}_t \leftarrow \frac{1}{n_1}\sum_{i=1}^{n_1}\nabla f(\mathbf{x}_t, \xi_i)$
5:     $\mathbf{H}_t \leftarrow \frac{1}{n_2}\sum_{i=1}^{n_2}\nabla^2 f(\mathbf{x}_t, \xi_i)$
6:     $\mathbf{\Delta}_t \leftarrow \arg\min_{\mathbf{\Delta}\in\mathbb{R}^d}\langle\mathbf{g}_t, \mathbf{\Delta}\rangle + \frac{1}{2}\langle\mathbf{\Delta}, \mathbf{H}_t\mathbf{\Delta}\rangle + \frac{M}{6}\|\mathbf{\Delta}\|^3$
7:     $\mathbf{x}_{t+1} \leftarrow \mathbf{x}_t + \mathbf{\Delta}_t$
8:     $t \leftarrow t + 1$
9: **end while**
10: **return** $\mathbf{x}_t$

---

**Definition 1** (Total sample complexity). *Given $\epsilon, \delta > 0$, the total sample complexity is the number of calls (queries) of stochastic gradient and stochastic Hessian along the iterations until reaching a point $\mathbf{x}$ that satisfies one of the following: (1) for high probability analysis: $F(\mathbf{x}) - F(\mathbf{x}^*) \leq \epsilon$ with probability at least $1 - \delta$; or (2) for analysis in expectation: $\mathbb{E}[F(\mathbf{x})] - F(\mathbf{x}^*) \leq \epsilon$.*

Our goal is to study the performance of stochastic cubic regularized Newton (SCRN) for objective functions that satisfy the gradient dominance property. Algorithm 1 describes the steps in SCRN. At each iteration $t$, we take batches of stochastic gradient vectors and Hessian matrices (lines 4 and 5) and then solve the following sub-problem to obtain $\mathbf{\Delta}_t$ (line 6):

$$\min_{\mathbf{\Delta}\in\mathbb{R}^d} m_t(\mathbf{\Delta}) := \langle\mathbf{g}_t, \mathbf{\Delta}\rangle + \frac{1}{2}\langle\mathbf{\Delta}, \mathbf{H}_t\mathbf{\Delta}\rangle + \frac{M}{6}\|\mathbf{\Delta}\|^3. \tag{3}$$

We assume that there is an oracle that returns a global solution for this sub-problem (This assumption will be relaxed subsequently, see Remark 6). Finally, we update $\mathbf{x}_t$ in line 7.

## 3 SCRN under gradient dominance property

We shall study the performance of SCRN for functions satisfying gradient dominance property, defined as follows.

**Assumption 2.** *Function $F(\mathbf{x})$ satisfies gradient dominance property when for every $\mathbf{x} \in \mathbb{R}^d$,*

$$F(\mathbf{x}) - F(\mathbf{x}^*) \leq \tau_F\|\nabla F(\mathbf{x})\|^\alpha, \tag{4}$$

*where $\mathbf{x}^* \in \arg\min_{\mathbf{x}} F(\mathbf{x})$, $\tau_F > 0$, and $\alpha \in [1, 2]$ are two constants.*

The case $\alpha = 2$ is often referred to as PL condition [28, 15]. In this paper, we consider all $\alpha$'s in the interval $[1, 2]$. The gradient dominance property holds for a large class of functions including sub-analytic functions, logarithm, and exponential functions, and semi-algebraic functions. These function classes cover some of the most common non-convex objectives used in practice (see related work in Section 1.1).

In the following lemma, we present a recursion inequality that captures the behaviour of the function $F(\mathbf{x}_t) - F(\mathbf{x}^*)$ at each iteration $t$ for SCRN under gradient dominance property.

**Lemma 1.** *Assume that function $F$ satisfies Assumption 1 (Lipschitz Hessian) and Assumption 2 (gradient dominance property) for $\alpha \geq 1$. Then the resulting update $\mathbf{x}_{t+1}$ in Algorithm 1 (line 7) after plugging in $\mathbf{\Delta}_t$, the solution of sub-problem in (3), satisfies the following:*

$$F(\mathbf{x}_{t+1}) - F(\mathbf{x}^*) \leq$$
$$C(F(\mathbf{x}_t) - F(\mathbf{x}_{t+1}))^{2\alpha/3} + C_g\|\nabla F(\mathbf{x}_t) - \mathbf{g}_t\|^\alpha + C_H\|\nabla^2 F(\mathbf{x}_t) - \mathbf{H}_t\|^{2\alpha}, \tag{5}$$

*where $C, C_g, C_H > 0$ are constants depending on $M, L_2$, and $\tau_F$, and defined in (34).*

Due to space limitations, all proofs are moved to the appendix.

In the following, we first provide an analysis in expectation of SCRN under gradient dominance property with $\alpha \in [1, 3/2]$. Next, we study the same algorithm for $\alpha \in (3/2, 2]$ using a high probability analysis.

### 3.1 SCRN under gradient dominance property with $\alpha \in [1, 3/2]$

We make the following assumption on the stochastic gradients and Hessians.

**Assumption 3.** *For a given $\alpha \in [1, 3/2]$ and for each query point $\mathbf{x} \in \mathbb{R}^d$:*

$$\mathbb{E}[\nabla f(\mathbf{x}, \xi)] = \nabla F(\mathbf{x}), \quad \mathbb{E}[\|\nabla f(\mathbf{x}, \xi) - \nabla F(\mathbf{x})\|_2^2] \leq \sigma_1^2, \tag{6}$$

$$\mathbb{E}[\nabla^2 f(\mathbf{x}, \xi)] = \nabla^2 F(\mathbf{x}), \quad \mathbb{E}[\|\nabla^2 f(\mathbf{x}, \xi) - \nabla^2 F(\mathbf{x})\|^{2\alpha}] \leq \sigma_{2,\alpha}^2, \tag{7}$$

*where $\sigma_1$ and $\sigma_{2,\alpha}$ are two constants.*

**Remark 1.** *The assumption $\mathbb{E}[\|\nabla^2 f(\mathbf{x}, \xi) - \nabla^2 F(\mathbf{x})\|^{2\alpha}] \leq \sigma_{2,\alpha}^2$ for $1 \leq \alpha \leq 3/2$ is slightly stronger than the usual assumption $\mathbb{E}[\|\nabla^2 f(\mathbf{x}, \xi) - \nabla^2 F(\mathbf{x})\|^2] \leq \sigma_2^2$. We need this assumption because of the specific form of the error of Hessian estimator in recursion inequality in (5). As a result of the assumption in (7), using a version of matrix moment inequality (See Lemma 4 in Appendix), we show that the dependency of the Hessian sample complexity in Theorem 1 on dimension $d$ is in the order of $\log d$.*

**Theorem 1.** *Let $F(\mathbf{x})$ satisfy Assumptions 1 and 2 for a given $\alpha$ and the stochastic gradient and Hessian satisfy Assumption 3 for the same $\alpha$. Moreover, assume that an exact solver for sub-problem (3) exists. Then Algorithm 1 outputs a point $\mathbf{x}_T$ such that $\mathbb{E}[F(\mathbf{x}_T)] - F(\mathbf{x}^*) \leq \epsilon$ after $T$ iterations, where*

*(i) if $\alpha \in [1, 3/2)$, $T = \mathcal{O}(\epsilon^{-\frac{3-2\alpha}{2\alpha}})$, with access to the following numbers of samples of the stochastic gradient and Hessian per iteration:*

$$n_1 \geq \frac{C_g^{2/\alpha}}{C^{6/\alpha}} \cdot \frac{4^{2/\alpha} \sigma_1^{2/\alpha}}{\epsilon^{2/\alpha}}, \quad n_2 \geq \frac{{C'_H}^{1/\alpha}}{C^{3/\alpha}} \cdot \frac{4^{1/\alpha} \sigma_{2,\alpha}^{2/\alpha}}{\epsilon^{1/\alpha}}, \tag{8}$$

*where $C'_H$ is defined in (38) and depends on $\log(d)$.*

*(ii) if $\alpha = 3/2$, $T = \mathcal{O}(\log(1/\epsilon))$ with the same numbers of samples per iteration as in (35).*

### 3.2 SCRN under gradient dominance property with $\alpha \in (3/2, 2]$

**Definition 2** (Bernstein's condition for matrices). *A zero-mean symmetric random matrix $\mathbf{X}$ satisfies the Bernstein condition with parameter $b > 0$ if*

$$\mathbb{E}[\mathbf{X}^k] \preccurlyeq \frac{1}{2} k! \, b^{k-2} \, \mathrm{Var}(\mathbf{X}), \quad for \ k = 3, 4, \dots \tag{9}$$

*where $\mathrm{Var}(\mathbf{X}) := \mathbb{E}[\mathbf{X}^2] - (\mathbb{E}[\mathbf{X}])^2$.*

**Assumption 4.** *We assume that the symmetric version of each centered gradient estimator $\mathbf{G}(\mathbf{x}, \xi) := \begin{bmatrix} \mathbf{0}_{1 \times 1} & \mathbf{g}(\mathbf{x}, \xi)^T \\ \mathbf{g}(\mathbf{x}, \xi) & \mathbf{0}_{d \times d} \end{bmatrix}$ where $\mathbf{g}(\mathbf{x}, \xi) := \nabla f(\mathbf{x}, \xi) - \nabla F(\mathbf{x})$ and each centered Hessian estimator $\mathbf{H}(\mathbf{x}, \xi) := \nabla^2 f(\mathbf{x}, \xi) - \nabla^2 F(\mathbf{x})$ satisfy Bernstein's condition (9) with parameters $M_1$ and $M_2$, respectively.*

**Remark 2.** *It is noteworthy that most previous work analyzing SCRN [33, 49, 36] assumed that centered gradient and centered Hessian estimators are bounded, i.e., $\|\nabla f(\mathbf{x}, \xi) - \nabla F(\mathbf{x})\|_2 \overset{a.s.}{\leq} M_1, \|\nabla^2 f(\mathbf{x}, \xi) - \nabla^2 F(\mathbf{x})\| \overset{a.s.}{\leq} M_2$. This is a stronger assumption than Assumption 4 as it implies Bernstein's condition (9) for $\mathbf{g}(\mathbf{x}, \xi)$ and $\mathbf{H}(\mathbf{x}, \xi)$.*

**Theorem 2.** *Suppose that $F(\mathbf{x})$ satisfies Assumptions 1, 2, the stochastic gradient and Hessian satisfy Assumption 3 (with $\alpha = 1$) and Assumption 4, and there exists an exact solver for sub-problem (3). Then, Algorithm 1, with probability $1 - \delta$, outputs a solution $\mathbf{x}_T$ such that $F(\mathbf{x}_T) - F(\mathbf{x}^*) \leq \epsilon$*

*after $T = \mathcal{O}\left(\log\left(\log(1/\epsilon)\right)\right)$ iterations with the following numbers of samples for the stochastic gradient and Hessian, respectively per iteration:*

$$n_1 \geq \frac{8}{3} \max\left(\frac{\tilde{C}^{1/\alpha} M_1}{\epsilon^{1/\alpha}}, \frac{\tilde{C}^{2/\alpha} \sigma_1^2}{\epsilon^{2/\alpha}}\right) \log\left(\frac{4(T+1)d}{\delta}\right), \tag{10}$$

$$n_2 \geq \frac{8}{3} \max\left(\frac{\tilde{C}^{1/(2\alpha)} M_2}{\epsilon^{1/(2\alpha)}}, \frac{\tilde{C}^{1/\alpha} \sigma_{2,1}^2}{\epsilon^{1/\alpha}}\right) \log\left(\frac{4(T+1)d}{\delta}\right), \tag{11}$$

*where $\tilde{C} = 1 + \frac{\tau_F}{2}\left(\frac{M+L_2+4}{2}\right)^\alpha$.*

**Remark 3.** *It is noteworthy that the assumptions in Theorem 2 suffice to obtain the sample complexities in Theorem 1 for $1 \leq \alpha \leq 3/2$ up to a logarithmic factor. However, the assumptions of Theorem 2 are stronger than those in Theorem 1.*

**Remark 4.** *Theorem 2 implies that the sample complexity of Algorithm 1 for $\alpha = 2$ is $\mathcal{O}\left(\log\log\log(1/\epsilon) \cdot \log(\log(1/\epsilon))/\epsilon\right)$.*

We summarized the sample complexity of SCRN in Theorems 1 and 2 and the best-known sample complexity of SGD under gradient dominance property in Table 1. The optimal sample complexity of SGD under gradient dominance property with $\alpha = 2$ is discussed in detail in [16, Section 5.2]. For the general case of $\alpha \in [1, 2]$, the sample complexity of SGD under gradient dominance property was derived in [10, Theorem 10]. This rate is achieved with a time-varying step-size. In the fourth column of Table 1, we provide the improvement of the sample complexity of SCRN with respect to that of SGD. The improvement is $\mathcal{O}\left(\epsilon^{-1/(2\alpha)}\right)$ for $\alpha \in [1, 3/2)$ and is $\tilde{\mathcal{O}}(\epsilon^{-2/\alpha+1})$ for $\alpha \in [3/2, 2]$, which are decreasing functions of $\alpha$. The largest improvement is for $\alpha = 1$ and is $\mathcal{O}(\epsilon^{-0.5})$.

**Remark 5.** *The special case $\alpha = 2$ generalizes strong convexity. For strongly convex objectives, the dependence of sample complexity of SGD on $\tau_F$ is $\mathcal{O}(\tau_F^2/\epsilon)$ [16, Corollary 2] while the sample complexity of SCRN is $\mathcal{O}(\tau_F^{7/4}/\epsilon)$. This is an improvement by a factor of $\tau_F^{1/4}$. In Appendix A.2.3, we provide simulation results comparing the performance of SCRN and SGD over synthetic functions by varying $\alpha$ and $\tau_F$.*

**Remark 6.** *In our analysis, we assumed that we have access to the exact solution of sub-problem in (3). Although no closed-form solution nor exact solver exists for this sub-problem, there are algorithms that approximate the exact solution with high probability [2, 4]. We emphasize that solving the sub-problem in (3) requires extra computation, but extra gradient and Hessian queries are not needed. In particular, Carmon and Duchi [4] proposed a perturbed GD-based algorithm that returns an approximate solution $\tilde{\boldsymbol{\Delta}}_t$ such that $m_t(\tilde{\boldsymbol{\Delta}}_t) \leq_{1-\delta'} m_t(\boldsymbol{\Delta}_t) + \epsilon'$ with $\mathcal{O}(\log(1/\delta')/\epsilon')$ iterations, for any given $\delta', \epsilon' > 0$.*

In Appendix A.2.2, we prove the following lemma:

**Lemma 2.** *Theorems 1 and 2 are still true if we were to use an inexact sub-solver which returned an approximate solution $\tilde{\boldsymbol{\Delta}}_t$ such that $\|\nabla m_t(\tilde{\boldsymbol{\Delta}}_t)\| \leq \epsilon^{1/\alpha}$ ($\epsilon^{1/\alpha}$-stationary point). Moreover, under some mild assumptions (same as those in [4]), a GD-based algorithm indeed returns such a solution in $\mathcal{O}(\epsilon^{-2/\alpha})$ iterations.*

**Remark 7.** *In the iterations of GD-based algorithm, instead of directly computing the Hessian matrix (which could be computationally expensive in high dimensions), we could compute Hessian-vector products by running Pearlmutter's algorithm [27]. Thus, the total computational complexity of evaluating gradients and Hessian vector products of SCRN is in the order of $\mathcal{O}(d\epsilon^{-3/\alpha})$ while the one of SGD is $\mathcal{O}(d\epsilon^{-4/\alpha+1})$, where $d$ is the dimension of $\mathbf{x}$. For $\alpha \in [1, 3/2)$, it can be seen that the computational complexity of SGD is less than the one of SCRN by a factor of $\mathcal{O}(\epsilon^{-1/(2\alpha)})$. For $\alpha \in [3/2, 2]$, this factor is $\tilde{\mathcal{O}}(\epsilon^{-(1-1/\alpha)})$.*

## 3.3 Further improvements

In our analysis of SCRN, we set the batch sizes such that the stochastic error terms $C_g\|\nabla F(\mathbf{x}_t) - \mathbf{g}_t\|^\alpha$ and $C_H\|\nabla^2 F(\mathbf{x}_t) - \mathbf{H}_t\|^{2\alpha}$ in (5) are in the order of $\epsilon$ (either in expectation or with high probability). However, for $1 \leq \alpha < 3/2$, it is just needed to make sure that the error terms at iteration $t$ are $\mathcal{O}(t^{-(2\alpha)/(3-2\alpha)})$, which equals the convergence rate of the function values $F(\mathbf{x}_t) - F(\mathbf{x}^*)$ (see Lemma 11 in Appendix A.3). Moreover, as stated in the following theorem, incorporating time-varying batch sizes in conjunction with variance reduction improves sample complexity results.

**Assumption 5.** *We assume that $f(\mathbf{x}, \xi)$ satisfies $L_1'$-average smoothness and $L_2'$-average Hessian Lipschitz continuity, i.e., $\mathbb{E}[\|\nabla f(\mathbf{x}, \xi) - \nabla f(\mathbf{y}, \xi)\|^2] \leq L_1'^2 \|\mathbf{x} - \mathbf{y}\|^2$ and $\mathbb{E}[\|\nabla^2 f(\mathbf{x}, \xi) - \nabla^2 f(\mathbf{y}, \xi)\|^2] \leq L_2'^2 \|\mathbf{x} - \mathbf{y}\|^2$ for all $\mathbf{x}, \mathbf{y} \in \mathbb{R}^d$.*

**Theorem 3.** *Suppose that $F(\mathbf{x})$ satisfies the gradient dominance property with $\alpha = 1$. Assume that Assumptions 1, 3, and 5 for $\alpha = 1$ hold. Then variance-reduced SCRN (See Algorithm 2 in Appendix A.3) achieves $\epsilon$-global stationary point in expectation by making $\mathcal{O}(\epsilon^{-2})$ stochastic gradients and $\mathcal{O}(\epsilon^{-1})$ stochastic Hessian on average, and the iteration complexity is $\mathcal{O}(\epsilon^{-1/2})$.*

We will see in the next section that there are a class of objective functions in RL setting that satisfies a weak version of gradient dominance with $\alpha = 1$ and Lipschitz gradient property and then we can take advantage of variance-reduced SCRN (See Remark 11).

## 4 SCRN under weak gradient dominance property in RL Setting

In this section, we showcase practical relevance of our result in Section 3.1 by applying SCRN to model-free RL. Specifically, in Theorem 4, we prove that as long as the expected return satisfies the weak version of gradient dominance property (Assumption 6), SCRN improves upon the best-known sample complexity of stochastic policy gradient (SPG) [41] by a factor of $\mathcal{O}(1/\sqrt{\epsilon})$ (see Appendix A.4.1 for the related work).

### 4.1 RL Setup

Consider a discrete Markov decision process (MDP) $\mathcal{M} = (\mathcal{S}, \mathcal{A}, P, R, \rho, \gamma)$, where $\mathcal{S}$ is the state space and $\mathcal{A}$ is the action space. $P(s'|s, a)$ denotes the probability of state transition from $s$ to $s'$ after taking action $a$ and $R(\cdot, \cdot) : \mathcal{S} \times \mathcal{A} \to [-R_{\max}, R_{\max}]$ is a bounded reward function, where $R_{\max}$ is a positive scalar. $\rho$ represents the initial distribution on state space $\mathcal{S}$ and $\gamma \in (0, 1)$ is the discount factor.

The parametric policy $\pi_\theta$ is a probability distribution over $\mathcal{S} \times \mathcal{A}$ with parameter $\theta \in \mathbb{R}^d$, and $\pi_\theta(a|s)$ denotes the probability of taking action $a$ at a given state $s$. Let $\tau = \{s_t, a_t\}_{t \geq 0} \sim p(\tau|\pi_\theta)$ be a trajectory generated by the policy $\pi_\theta$, where $p(\tau|\pi_\theta) := \rho(s_0) \prod_{t=0}^{\infty} \pi_\theta(a_t|s_t) P(s_{t+1}|s_t, a_t)$. The expected return of $\pi$ is defined as $J(\pi_\theta) := \mathbb{E}_{\tau \sim p(\cdot|\pi_\theta)} \left[\sum_{t=0}^{\infty} \gamma^t R(s_t, a_t)\right]$. In the sequel, we consider a set of parameterized policies $\{\pi_\theta : \theta \in \mathbb{R}^d\}$, with the assumption that $\pi_\theta$ is differentiable with respect to $\theta$. For ease of presentation, we denote $J(\pi_\theta)$ by $J(\theta)$.

The goal of policy-based RL is to find $\theta^* = \arg\max_\theta J(\theta)$. However, in many cases, $J(\theta)$ is a non-concave function and instead we settle for obtaining an $\epsilon$-FOSP, $\hat\theta$, such that $\|\nabla J(\hat\theta)\| \leq \epsilon$. It can be shown that: $\nabla J(\theta) = \mathbb{E}\left[\sum_{h=0}^{\infty} \left(\sum_{t=h}^{\infty} \gamma^t R(s_t, a_t)\right) \nabla \log \pi_\theta(a_h|s_h)\right]$. In practice, the full gradient cannot be computed due to the infinite horizon length. Instead, it is commonly truncated to a length H horizon as follows. $\nabla J_{\mathsf{H}}(\theta) = \mathbb{E}[\sum_{h=0}^{\mathsf{H}-1} \Psi_h(\tau) \nabla \log \pi_\theta(a_h|s_h)]$, where $\Psi_h(\tau) = \sum_{t=h}^{\mathsf{H}-1} \gamma^t R(s_t, a_t)$.

Assume that we sample $m$ trajectories $\tau^i = \{s_t^i, a_t^i\}_{t \geq 0}$, $1 \leq i \leq m$, and then compute $\hat\nabla_m J(\theta) = \frac{1}{m} \sum_{i=1}^{m} \sum_{h=0}^{\mathsf{H}-1} \Psi_h(\tau^i) \nabla \log \pi_\theta(a_h^i|s_h^i)$, which is an unbiased estimator for $\nabla J_{\mathsf{H}}(\theta)$. The vanilla SPG method is based on the following update: $\theta \leftarrow \theta + \eta \hat\nabla_m J(\theta)$, where $\eta$ is the learning rate.

It can be shown that the Hessian matrix of $J_{\mathsf{H}}(\theta)$ can be obtained as follows [29, Appendix 7.2]: $\nabla^2 J_{\mathsf{H}}(\theta) = \mathbb{E}[\nabla \Phi(\theta; \tau) \nabla \log p(\tau|\pi_\theta)^T + \nabla^2 \Phi(\theta; \tau)]$, where $\Phi(\theta; \tau) = \sum_{h=0}^{\mathsf{H}-1} \sum_{t=h}^{\mathsf{H}-1} \gamma^t r(s_t, a_t) \log \pi_\theta(a_h|s_h)$. As a result, for trajectories $\tau^i = \{s_t^i, a_t^i\}_{t \geq 0}$, $1 \leq i \leq m$, $\hat\nabla_m^2 J(\theta) = \frac{1}{m} \sum_{i=1}^{m} \nabla \Phi(\theta; \tau^i) \nabla \log p(\tau^i|\pi_\theta)^T + \nabla^2 \Phi(\theta; \tau^i)$ is an unbiased estimator of Hessian matrix $\nabla^2 J_{\mathsf{H}}(\theta)$.

### 4.2 Sample Complexity of SCRN

In our analysis, we consider the recently introduced relaxed weak gradient dominance property with $\alpha = 1$ [41].

**Assumption 6** (Weak gradient dominance property with $\alpha = 1$). *$J$ satisfies the weak gradient dominance property if for all $\theta \in \mathbb{R}^d$, there exist $\tau_J > 0$ and $\epsilon' > 0$ such that*

$$\epsilon' + \tau_J \|\nabla J(\theta)\| \geq J^* - J(\theta), \tag{12}$$

*where $J^* := \max_\theta J(\theta)$.*

**Remark 8.** *Two commonly made assumptions in the literature: non-degenerate Fisher matrix [20, 9] and transferred compatible function approximation error [35, 1] imply Assumption 6. See Appendix A.4.1 for more details.*

Further, we also make Lipschitz and smooth policy (LS) assumptions which are widely adopted in the analysis of vanilla policy gradient (PG) [46] as well as variance-reduced PG methods, e.g. in [29].

**Assumption 7** (LS)**.** *There exist constants $G_1, G_2 > 0$ such that for every state $s \in \mathcal{S}$, the gradient and Hessian of $\log \pi_\theta(\cdot|s)$ satisfy $\|\nabla_\theta \log \pi_\theta(a|s)\| \leq G_1$ and $\|\nabla^2_\theta \log \pi_\theta(a|s)\| \leq G_2$.*

**Lemma 3.** *Under Assumption 7, we have $\|\nabla J(\theta) - \nabla J_{\mathsf{H}}(\theta)\| \leq D_g \gamma^{\mathsf{H}}$ and $\|\nabla^2 J(\theta) - \nabla^2 J_{\mathsf{H}}(\theta)\| \leq D_H \gamma^{\mathsf{H}}$, where $D_g = \frac{G_1 R_{\max}}{1-\gamma} \sqrt{\frac{1}{1-\gamma} + \mathsf{H}}$ and $D_H = \frac{R_{\max}(G_2 + G_1^2)}{1-\gamma} \left( \mathsf{H} + \frac{1}{1-\gamma} \right).$*

**Assumption 8** (Lipschitz Hessian)**.** *There exists a constant $\bar{L}_2$ such that the Hessian of $\log \pi_\theta(a|s)$ satisfies*

$$\|\nabla^2 \log \pi_\theta(a|s) - \nabla^2 \log \pi_{\theta'}(a|s)\| \leq \bar{L}_2 \|\theta - \theta'\|. \tag{13}$$

The Lipschitz Hessian assumption is commonly used to find SOSP in policy gradient algorithms [40]. For the Gaussian policy (136), $\nabla^2 \log \pi_\theta(a|s)$ reduces to the matrix $-\phi(s)\phi(s)^T/\sigma^2$, which is a constant function of $\theta$ and thus satisfies condition (13). Soft-max policy also satisfies this assumption (See appendix A.4.4).

**Theorem 4.** *For a policy $\pi_\theta$ satisfying Assumptions 7, 8, and the corresponding objective function $J(\theta)$ satisfying Assumption 6, SCRN outputs the solution $\theta_T$ such that $J^* - \mathbb{E}[J(\theta_T)] \leq \epsilon + \epsilon'$ and the sample complexity (the number of observed state-action pairs) is: $T \times m \times \mathsf{H} = \tilde{\mathcal{O}}(\epsilon^{-2.5})$ for $\epsilon' = 0$ and $T \times m \times \mathsf{H} = \tilde{\mathcal{O}}(\epsilon^{-0.5}\epsilon'^{-2})$ for $\epsilon' > 0$.*

**Remark 9.** *Under weak gradient dominance property with $\alpha = 1$ (Assumption 6), it has been shown that the sample complexity of SPG is $\tilde{\mathcal{O}}(\epsilon^{-3})$ in case of $\epsilon' = 0$ and $\tilde{\mathcal{O}}(\epsilon^{-1}\epsilon'^{-2})$ in case of $\epsilon' > 0$ [41, Theorem C.1]. Therefore, SCRN improves upon the best-known sample complexity of SPG in both cases $\epsilon' = 0$ and $\epsilon' > 0$ by a factor of $\mathcal{O}(\epsilon^{-0.5})$.*

**Remark 10.** *Having access to exact gradient and Hessian (the deterministic case), under Assumption 6, PG algorithm achieves global convergence (i.e., $J^* - J(\theta_T) \leq \epsilon$) with $\tilde{\mathcal{O}}(\epsilon^{-1})$ iterations [41] while CRN requires $\tilde{\mathcal{O}}(\epsilon^{-0.5})$ iterations.*

**Remark 11.** *Under the same assumptions as in Theorem 4 and Assumption 5 for $\alpha = 1$ and bounded variance of importance sampling weights (See Assumption 15), a variance-reduced version of SCRN (See Algorithm 3) achieves global convergence (i.e., $J^* - \mathbb{E}[J(\theta_T)] \leq \epsilon$) with a sample complexity of $\tilde{\mathcal{O}}(\epsilon^{-2})$. See Appendix A.4.6 for a proof.*

## 5 Experiments

In this section, we evaluate the performance of SCRN in the two following RL settings. First, we consider grid world environments with finite state and action spaces, and next some robotic control tasks with continuous state and action spaces. The details of the implementations for all methods and some additional experiments appear in the appendix and the codes are available in the supplementary material.

**Environments with finite state and action spaces:** We consider two grid world environments in our experiments: cliff walking [31, Example 6.6], and random mazes [51]. In cliff walking, the agent's aim is to reach a goal state from a start state, avoiding a region of cells called "cliff". The episode is terminated if the agent enters the cliff region, or the number of steps exceeds 100 without reaching the goal. Moreover, we consider a soft-max tabular policy in the experiments of this part. Indeed, it has been shown that a variant of gradient dominance property with $\alpha = 1$ holds for soft-max tabular policy in environments with finite state and action spaces [22].

We compare SCRN with two existing first-order methods: vanilla SPG and REINFORCE [37]. For the first-order methods, we use a time-varying learning rate and tune the parameters. To improve the performance of the first-order methods, we also add an entropy regularization term to the reward function.

In Fig. 1, the average length of paths traversed by the agent and the average episode return are depicted against the number of episodes for each method. The results are averaged over 64 instances

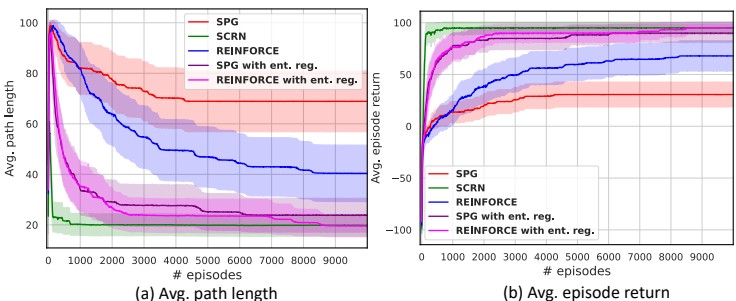

(a) Avg. path length        (b) Avg. episode return

Figure 1: Comparison of SCRN with first-order methods in cliff walking environment. The percentages of successful instances for SPG, SCRN, REINFORCE, SPG with entropy regularization, and REINFORCE with entropy regularization are $32.8\%$, $100\%$, $54.7\%$, $100\%$, and $92.2\%$, respectively.

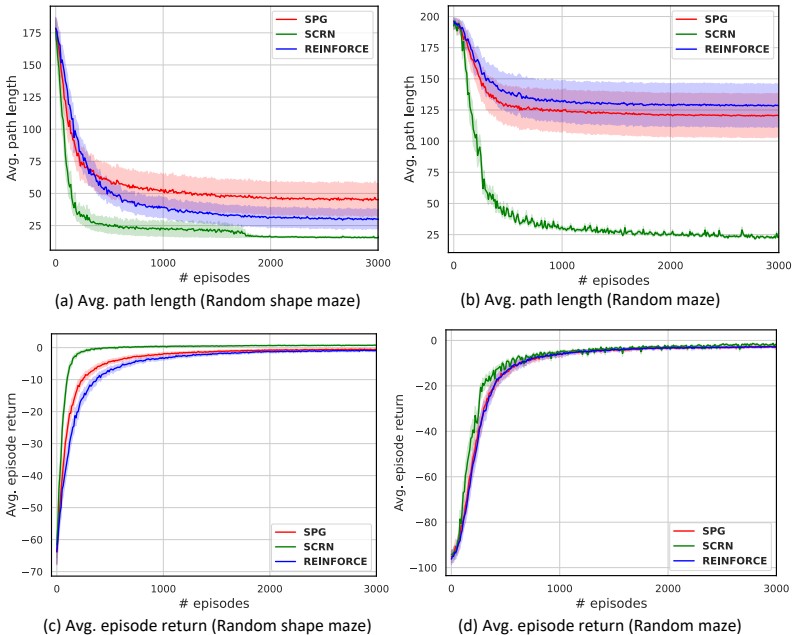

(a) Avg. path length (Random shape maze)    (b) Avg. path length (Random maze)

(c) Avg. episode return (Random shape maze)    (d) Avg. episode return (Random maze)

Figure 2: Comparison of SCRN with first-order methods in maze environments. In random shape maze, the percentages of successful instances for SPG, SCRN, and REINFORCE are $86\%$, $100\%$, and $95.3\%$, respectively. In random maze, the percentages of successful instances for SPG, SCRN, and REINFORCE are $45.3\%$, $97\%$, and $40.6\%$ respectively.

and the shaded region shows the $90\%$ confidence interval. As can be seen in Fig. 1 (a), all the algorithms have a phase at the beginning during which the agent falls off the cliff in most episodes and the average length of paths are small. Then, the agent learns to avoid the cliff but could still not reach the goal in most cases. Finally, it finds a path to the goal and tries to reduce the path length. Note that SCRN finds the path very quickly and significantly outperforms the other algorithms. In fact, SPG and REINFORCE get stuck in the start state for some period of time, while SCRN easily escapes the flat plateau (for more details, please see the demonstrations in the supplementary file). The performance of SPG and REINFORCE improves with entropy regularization, but SCRN still outperforms them. Moreover, SPG and REINFORCE fail to reach the goal in some instances while SCRN is successful in almost all instances. In the captions of figures, for each algorithm, we also provide the percentage of instances in which the agent reached the goal. Please note that these percentages are obtained based on the parameters from the last update of each algorithm.

Additionally, we studied the performance of the three aforementioned algorithms on a random maze and a random shape maze [51]. In the random shape maze, random shape blocks are placed on a grid and the agent tries to reach the goal state finding the shortest path. As shown in Fig. 2, SCRN again

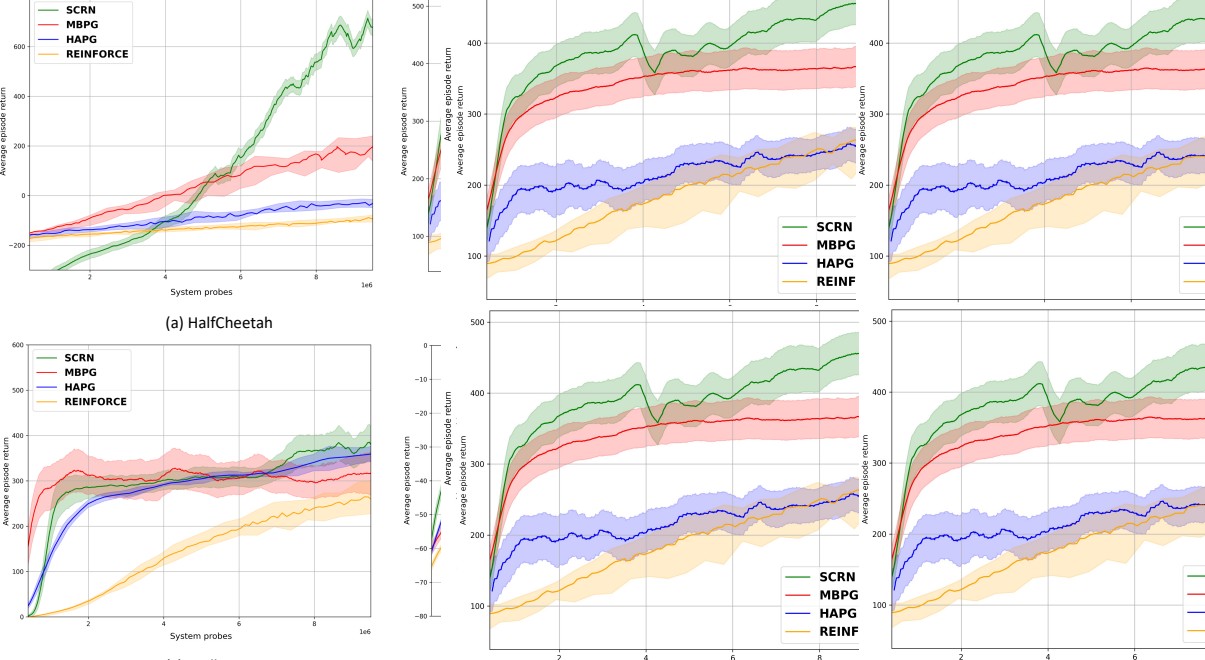

Figure 3: Comparison of SCRN with REINFORCE and variance-reduced SPG methods in MuJoCo environments.

outperforms the first-order methods in both environments. In Appendix A.5, we also provided results for SPG and REINFROCE with entropy regularization which have slightly better performance.

**Environments with continuous state and action spaces:** We consider the following control tasks in MuJoCo simulator [32]: Walker, Humanoid, Reacher, and HalfCheetah. We compare SCRN with first-order methods such as REINFORCE, and two state-of-the-art representatives of variance-reduced PG methods, HAPG [29] and MBPG [13], both with guaranteed convergence to $\epsilon$-FOSP in general non-convex settings. HAPG uses second order information (Hessian vector products) for variance reduction and MBPG is a recent work based on STORM, a batch-free state-of-the-art variance reduction approach [8].

We report average episode return against system probes as our performance measure. That is, the number of observed state-action pairs (see Fig. 3). At each point, we run the trained policy 10 times and compute the empirical estimate of the mean and the $90\%$ confidence interval of the episode return. As seen in Fig. 3, SCRN outperforms the other methods, especially in more complex environments such as HalfCheetah and Humanoid. In Appendix A.5, we also provided results for variance-reduced SCRN which improves upon SCRN in Humanoid and Reacher environments.

## 6 Conclusion

We studied the performance of SCRN for objectives satisfying the gradient dominance property for $1 \leq \alpha \leq 2$, which holds in various machine learning applications. We showed that SCRN improves the best-known sample complexity of SGD. The largest improvement is in the case of $\alpha = 1$. Moreover, for $\alpha = 1$, the average sample complexity of SCRN can be reduced to $\mathcal{O}(\epsilon^{-2})$ by utilizing a variance reduction method with time-varying batch sizes. A weak version of gradient dominance for $\alpha = 1$ is satisfied in some policy-based RL settings. In the RL setting, we showed that SCRN achieves the same improvement over SPG under the weak version of gradient dominance property for $\alpha = 1$.

## 7 Acknowledgement

The authors would like to thank Mohammadsadegh Khorasani for conducting the experiments for continuous state-action environments. This research is partially supported by National Centre of Competence in Research (NCCR), grant agreement no. 51NF40_180545.

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
