# OpenReview forum: "Stochastic Second-Order Methods Improve Best-Known Sample Complexity of SGD for Gradient-Dominated Functions"
_NeurIPS.cc/2022/Conference — NeurIPS 2022 Accept_

### Official Review · Reviewer_jWxP · 2022-06-24

**Rating:** 6
**Confidence:** 4
**Soundness:** 4 excellent
**Presentation:** 3 good
**Contribution:** 2 fair

**Summary:**

In this paper the authors considers the general non-convex stochastic programming problem with gradient dominance property. To solve this problem, a stochastic subsampled Nestrov's Cubic Regularized Newton Method is proposed. In general, an improved sample complexity is derived compared to the SGD method. (In terms of dependence on $\epsilon$, variance reduced method does not perform better than SGD for gradient dominant problems.)

In particular, for gradient dominant problems with $\alpha = 1$, a sample complexity of $O(\epsilon^{-2.5})$ is obtained. This special case is then connected to the cumulative reward optimization in RL problem, where the policy is parameterized in a soft-max form. A SARAH (or SPIDER) type variance reduced variant of the method is proposed and discussed for this case.

Overall, the theoretical result in this paper is solid and the numerical experiment of this paper is complete. The reviewer believes that the authors does complete a missing part of discussion in the stochastic cubic regularized method. However, despite the complicated technical discussion, the result is somehow (conceptually) straightforward and can be expected. Therefore, the reviewer would like to provide an overall rating of weak accept.

**Questions:**

Minor comment #1: In Definition 2, line 160, the definition of $E[X^2]$ and $(E[X])^2$ is not clear for matrix input. If the squares are applied element-wisely, the authors should clearly state that.

Minor comment #2: In Theorem 2, line 170, ``($\alpha=1$)'' does not seems correct. The discussion is made for $\alpha\in(1.5,2]$. The same  issue also exists in the appendix.

Minor comment #3: In Remark 6, I think the authors should emphasize that solving the subproblems requires extra computation, but it does not require extra samples. Therefore, no matter how the subproblem is solved, it will not worsen the theoretical result obtained in Theorem 1 & 2.

Minor comment #4: In line 212-213, the authors state that ``Moreover, as stated in the following theorem, incorporating time varying batch sizes in conjunction with variance reduction improves sample complexity results.'' However, from the appendix and Theorem 1, it seems that both the variance reduced version and the naive subsampled version both achieve the sample complexity of $O(\epsilon^{-2.5})$. The reviewer is confused where the improvement in sample complexity is.

Minor comment #5: The statement of Theorem 3 is confusing. First, the iteration complexity is missing in the theorem. Second, the term "on average'' is confusing. It will be better to say the "average batchsizes per iteration is ...''

Minor comment #6: Is the weak gradient dominant property really proved for soft-max parameterized policy for nontrivial $\epsilon'$? The reviewer is not sure about this.

**Limitations:**

NA. This is a theoretical paper.

**Strengths And Weaknesses:**

Strength #1: The paper complete a missing part of discussion in the stochastic cubic regularized method when the objective function is gradient dominant.

Strength #2: The theoretical result is solid, and the reviewer believe the derivation are correct.

Strength #3: The numerical experiment are complete.

Weakness #1: The methodology is not quite novel, from my personal point of view.

---

> ### Author Response · Authors · 2022-08-02
> **Response to Reviewer jWxP**
>
> > Weakness 1: The methodology is not quite novel, from my personal point of view.
>
> Some novelties of our methodology are: 1- We provided a stochastic analysis, in which
> separating the estimation error terms in the additive form in the recursion is crucial to our derivations (Lemma 1). Generic proof techniques in the deterministic case are not directly applicable. 2- One possible approach to achieve $\epsilon$-global stationary point, is to use fixed batch-sizes in all the iterations such that the error terms are bounded in the order of $\epsilon$. Instead, we showed that it suffices to make sure that the error terms are decaying at the same rate as the function values (Lemma 10 in Appendix A.3). Combining this idea with a variance-reduction technique yields a novel method which reduces the sample complexity to $O(\epsilon^{-2})$ for $\alpha=1$. 3- We proposed an expectation analysis for SCRN under gradient dominance property with $1\leq \alpha\leq 3/2$ which requires weaker assumptions than the bounded centered gradient and Hessian estimators which are commonly assumed for analyzing SCRN in general non-convex setting in the literature. Under gradient dominance condition with $1\leq \alpha\leq 3/2$, dependency of Hessian sample complexity on dimension is reduced from $poly(d)$ to $\log d$ ($d$ is the dimension of the ambient space) by using a version of matrix moment inequality (See Lemma 3 in Appendix).
>
> > Questions: Minor comment 1: In Definition 2, line 160, the definition of E[X2] and (E[X])2 is not clear for matrix input. If the squares are applied element-wisely, the authors should clearly state that.
>
> As $X$ is a squared matrix, $X^{2}$ means the matrix multiplication of $X\times X$. The expectation is taken component-wise and $(\mathbb{E}[X])^2$ is the matrix multiplication of $\mathbb{E}[X]$ by itself.
>
> > Minor comment 2: In Theorem 2, line 170, ``($\alpha=1$)'' does not seems correct. The discussion is made for $\alpha\in(1.5,2]$. The same issue also exists in the appendix.
>
> Assumption 3 with $\alpha=1$ corresponds to the commonly used assumption of bounded variance for gradient and Hessian, which is needed for Theorem 2.
>
> > Minor comment 3: In Remark 6, I think the authors should emphasize that solving the subproblems requires extra computation, but it does not require extra samples. Therefore, no matter how the subproblem is solved, it will not worsen the theoretical result obtained in Theorem 1 and 2.
>
> Thank you for your insightful comment on Remark 6. We added this explanation to Remark 6 (lines 195-197 in the revised version).
>
> > Minor comment 4: In line 212-213, the authors state that ``Moreover, as stated in the following theorem, incorporating time varying batch sizes in conjunction with variance reduction improves sample complexity results.'' However, from the appendix and Theorem 1, it seems that both the variance reduced version and the naive subsampled version both achieve the sample complexity of O($\epsilon^{-2.5}$). The reviewer is confused where the improvement in sample complexity is.
>
> The proposed variance reduced SCRN achieves sample complexity of $O(\epsilon^{-2})$ on average for $\alpha=1$ and improves the sample complexity of $O(\epsilon^{-2.5})$ for the sub-sampled version of SCRN.
>
> > Minor comment 5: The statement of Theorem 3 is confusing. First, the iteration complexity is missing in the theorem. Second, the term "on average'' is confusing. It will be better to say the "average batchsizes per iteration is ...''
>
> The iteration complexity is $T=O(1/\sqrt{\epsilon})$ and we added it to the statement of Theorem 3. Regarding average batch-sizes per iteration, the number of the queries of gradient and Hessian estimators depend on the norm of $||\mathbf{\Delta}_{t-1}||$ in the previous iteration. Thus, we only provide the average total sample complexity.
>
> > Minor comment 6: Is the weak gradient dominant property really proved for soft-max parameterized policy for nontrivial $\epsilon'$? The reviewer is not sure about this.
>
> Non-uniform version of gradient dominant property with $\alpha=1$ has been shown in reference [22]. In particular, $\tau_J$ depends on $\theta$ in the non-uniform version. Moreover, in reference [9], the authors claimed that $\epsilon'=0$ for softmax tabular policy.

---

> ### Author Response · Authors · 2022-08-08
> **Author-Reviewer discussion period**
>
> Dear Review jWxP, as the discussion period is ending soon, we would like to know if we have addressed your comment about the novelty of the methodology.  Do you have any further questions that we can clarify?

---

> > ### Comment · Reviewer_jWxP · 2022-08-08
> > **Response to comment**
> >
> > My main concern is only about the novelty and importance of the paper. Though the authors argued that they did various analysis in the stochastic setting. For my own point of view, I feel these result standard and can be expected. Therefore, I will remain the rating of 6 for this submission.

---

### Official Review · Reviewer_GSWi · 2022-07-08

**Rating:** 7
**Confidence:** 4
**Soundness:** 4 excellent
**Presentation:** 3 good
**Contribution:** 3 good

**Summary:**

This paper analyzes the convergence of the stochastic cubic regularized Newton (SCRN) methods for $\alpha$-gradient-dominant functions. It shows that SCRN consistently outperforms stochastic gradient descent (SGD) for all values of $\alpha \in [1,2]$. When $\alpha = 1$ as a special case in a reinforcement learning (RL) setting, this also improves the sample complexity compared to the stochastic policy gradient (SPG). Experiments are provided to support the claims in the RL settings. Finally, a variance reduced version of SCRN is considered and its sample complexity is improved for $\alpha = 1$ compared to the original SCRN.

**Questions:**

1. The paper is well positioned in the literature. The paper asks, does SCRN improve upon SGD under gradient dominance property, which is an important research question. However, why do we care about gradient dominance property ? I think the authors could bring more motivation on this in the introduction. In the SGD literature, gradient dominance property helps to get faster global optimum convergence results (Karimi et al., Khaled et al.). What are the SCRN convergence results without gradient dominance property in the previous literature ?

Also, it might be good to elaborate line 52-53: CRN outperforms GD under gradient dominance property for all $\alpha \in [1,2]$. What are the numbers of iterations in these cases ?

2. Remark 10: The variance reduced version of SCRN in Algorithm 2 applied in RL does not use a distribution shift term here (like equation (7) in [1]), which is a common technique in the variance reduced policy gradient methods. This will induce a bias for the estimates of ${\bf v}_t$ and ${\bf U}_t$ in Algorithm 2. The proof in Appendix A.4.6 does not deal with this bias. Thus I have a doubt on the results of Remark 10.

3. Line 316: we also provide the percentage of instances in which the agent reached the goal. Is this percentage obtained by using the last iterate of the algorithm ?

4. What's the difference between SPG and REINFORCE ? They seem the same for me.

5. In the experiments, how did you solve the sub-problem for each iteration ? Did you use the inexact cubic sub-solver presented in Appendix A.2.2 ?

6. It would be interesting to see experiments for variance reduced SCRN as well.

[1] Feihu Huang, Shangqian Gao, Jian Pei, and Heng Huang. Momentum-based policy gradient methods. In International Conference on Machine Learning, pages 4422–4433. PMLR, 2020.


Typo:
Line 650: we can get the desired result in Theorem "2"

**Limitations:**

The authors have adequately addressed the limitations of their work.

**Strengths And Weaknesses:**

The major claims of the paper (Theorem 1 & 2) seem solid to me. Through Table 1, it is clear that SCRN provably beats SGD for all $\alpha \in [1,2]$.

The paper not only has the main results Theorem 1 & 2, but also develops many other interesting works, such as the improved sample complexity of SPG in RL with experiments and the sample complexity of the variance reduced version of SCRN when $\alpha = 1$. This makes the paper strong and complete.

However, I found Theorem 1 & 2 to be quite hard to follow. For instance, my knowledge on Bernstein's condition is limited. Assumption 4 seems complex and not natural to me, what is the interpretation of this assumption ? From Remark 2, I understood that this is a weaker assumption as compared to the most previous SCRN analysis. A part from this, what role does it play in the proof of Theorem 2 ? Did the previous SCRN analysis also use Assumption 1 & 3 ? In other words, are Assumption 1 and 3 common in SCRN analysis ?

---

> ### Author Response · Authors · 2022-08-02
> **Response to Reviewer GSWi**
>
> > However, I found Theorem 1 and 2 to be quite hard to follow. For instance, my knowledge on Bernstein's condition is limited. Assumption 4 seems complex and not natural to me, what is the interpretation of this assumption?
>
> The Bernstein condition for random matrices yields the sub-exponential condition in reference [34, Chap. 6] of our paper. The sub-exponential condition is an extension of the definition of sub-exponential random variables to random symmetric matrices. In particular, the sub-exponential condition for a zero-mean symmetric random matrix guarantees that the tail probability of the operator norm of that matrix decreases at least exponentially.
>
> > From Remark 2, I understood that this is a weaker assumption as compared to the most previous SCRN analysis. A part from this, what role does it play in the proof of Theorem 2 ? Did the previous SCRN analysis also use Assumption 1 and 3 ? In other words, are Assumption 1 and 3 common in SCRN analysis ?
>
> We used Bernstein’s inequality in order to control the error terms of gradient and Hessian estimators in the recursive inequality in Lemma 1. Moreover, nearly all previous works on analyzing SCRN for the general non-convex setting make Assumption 1. As we mentioned in Remark 2, these works also assume bounded centered gradient and centered Hessian estimators, which are stronger assumptions than Assumption 3.
>
> > Questions: 1. The paper is well positioned in the literature. The paper asks, does SCRN improve upon SGD under gradient dominance property, which is an important research question. However, why do we care about gradient dominance property ? I think the authors could bring more motivation on this in the introduction. In the SGD literature, gradient dominance property helps to get faster global optimum convergence results (Karimi et al., Khaled et al.). What are the SCRN convergence results without gradient dominance property in the previous literature ?
>
> In the submitted version, we discussed some applications of gradient dominance property in the related work. Furthermore, we discussed SCRN convergence results without gradient dominance property in the related work in the part "Variants of cubic regularized Newton
> method”.
>
> > Also, it might be good to elaborate line 52-53: CRN outperforms GD under gradient dominance property for all $\alpha\in[1,2]$. What are the numbers of iterations in these cases?
>
> CRN: for $\alpha\in [1,3/2)$ the number of iterations is $O(1/\epsilon^{3/(2\alpha)-1})$ , for $\alpha=3/2$ the number of iterations is $O(\log(1/\epsilon))$, and for $\alpha\in (3/2,2]$ the number of iterations is $O(\log\log(1/\epsilon))$.
>
> GD:
> for $\alpha\in [1,2)$: the number of iterations is $O(1/\epsilon^{2/\alpha-1})$ and for $\alpha=2$ the number of iterations is $O(\log(1/\epsilon))$. We added these in a footnote on page 2 of the revised version.
>
> >  2-	Remark 10: The variance reduced version of SCRN in Algorithm 2 applied in RL does not use a distribution shift term here (like equation (7) in [1]), which is a common technique in the variance reduced policy gradient methods. This will induce a bias for the estimates of vt and Ut in Algorithm 2. The proof in Appendix A.4.6 does not deal with this bias. Thus I have a doubt on the results of Remark 10.
>
> Thanks for the invaluable comment. With an extra assumption on the boundedness of the variance of importance sampling (IS) weights, IS weights can be used to mitigate the bias in the estimators of gradient and Hessian. We added this assumption to the statement of Remark 10 and revised the proof.
>
> > 3-	Line 316: we also provide the percentage of instances in which the agent reached the goal. Is this percentage obtained by using the last iterate of the algorithm ?
>
> Yes, this percentage is obtained based on the last iterate of the algorithm. We clarified this in the revised version (lines 321-322).
>
> > 4-	What's the difference between SPG and REINFORCE? They seem the same for me.
>
> The difference is in using two different forms of unbiased estimator of the gradient of the value function. We defined vanilla SPG updates in lines 248 to 250 of the submitted version. The update rule of REINFORCE is given in reference [37] of our paper.
>
> > 5-	In the experiments, how did you solve the sub-problem for each iteration ? Did you use the inexact cubic sub-solver presented in Appendix A.2.2?
>
> Yes, we used the inexact sub-solver introduced in Appendix A.2.2.
>
> > 6-	It would be interesting to see experiments for variance reduced SCRN as well.
>
> We performed experiments for variance-reduced SCRN in control tasks and added the results in Appendix A.5. In Humanoid, Reacher, and Walker environments, the variance-reduced SCRN achieves the average episode return of about $450$, $-10$, and $400$ with lower system probes compared with SCRN, respectively.
>
> > Typo: Line 650: we can get the desired result in Theorem "2"
>
> Thank you for your comment. We corrected it.

---

> ### Author Response · Authors · 2022-08-08
> **Author-Reviewer discussion period**
>
> Dear Review GSWi, as the discussion period is ending soon, we would like to know if we have addressed your concerns about Remark 10.  Do you have any further questions that we can clarify?

---

> > ### Comment · Reviewer_GSWi · 2022-08-08
> > **Thank you and further comments on Remark 10**
> >
> > Thank you for responding comprehensively to all my comments, especially for the remind of some discussions in the related work that I missed in my previous review.
> >
> > As for my concern about Remark 10, this is the answer that I expected. When estimating the gradients and the Hessians, you need to add IS for both of them. Intuitively, this might make the order of $\mathcal{O}(\frac{1}{1-\gamma})$ high. Also, you might need to add a RL version of the pseudocode in the appendix in the revised version as well.
> >
> > Since I do not see the proof with this extra assumption and it might have other difficulties hidden in the proof, I will remain my score for this submission.

---

> > > ### Author Response · Authors · 2022-08-08
> > > **Thank you for checking the rebuttal**
> > >
> > > Thanks a lot for checking the rebuttal. Regarding the RL version of pseudocode and proof of the revised version of Remark 10, we added them in the revised paper in Appendix A.4.6 (lines 932-977). Please find the revised version (including the appendix) in the file with the name “Full_paper.pdf” in the supplementary material.

---

> > > > ### Comment · Reviewer_GSWi · 2022-08-08
> > > > **I will defend my score for the paper.**
> > > >
> > > > Thanks for the great effort of fixing the proofs and updating the paper accordingly. Now the results are complete and sound.
> > > >
> > > > One additional suggestion for the results in RL setting: In RL, the order of $\mathcal{O}(\frac{1}{1-\gamma})$ matters. It is arguable that this order can be as important as the one of $\epsilon$. It would be nice to provide as well the order of $\mathcal{O}(\frac{1}{1-\gamma})$ for the sample complexity results in the final version.
> > > >
> > > > When I initially gave my rating of 7, I believed already that the issue in the proof for Remark 10 could be fixed. Thus I will still remain my score. However, I will defend my score for the paper in the Reviewer-Metareviewer Discussion period.

---

> > > > > ### Author Response · Authors · 2022-08-08
> > > > > **Thank you for your comment**
> > > > >
> > > > > Thank you for your comment. The dependency of sample complexity for both SCRN and VR-SCRN in terms of $\frac{1}{1-\gamma}$ and the horizon $\mathsf{H}$ will be added in the final version. As noted by the reviewer, the importance sampling term makes an extra coefficient $\mathsf{H}$ (which is in order of $\frac{1}{1-\gamma}$ in our analysis. See line 948 in appendix A.4.6) for the errors of gradient and Hessian estimators. See Equations (167) and (168) of our proof of Remark 10 in which upper bounds on the gradient and Hessian error terms depend on $C_{w}$ which is proportional to horizon $\mathsf{H}$.

---

### Official Review · Reviewer_UAyg · 2022-07-10

**Rating:** 4
**Confidence:** 3
**Soundness:** 2 fair
**Presentation:** 2 fair
**Contribution:** 3 good

**Summary:**

The authors show sample complexity bounds for stochastic cubic regularized Newton (SCRN) on gradient dominated functions. Their bounds improve on upper bounds for SGD. The authors present experiments on RL problems using SCRN.

**Questions:**

Is it not already known that SCRN performs empirically well on RL problems? I am not an expert on optimization methods for RL but it seems surprising. Could you perhaps suggest why no other papers have tested SCRN on RL problems.

Does your implementation work poorly on non-RL problems?

How are the proof techniques different from standard techniques [15,25]?

Have you experimented with variance reduced SCRN? How does it compare with SCRN?

Are all the problems you test on twice-differentiable? Is that typical in RL?

The algorithm presented by the authors requires three highly problem-dependent hyperparameters. Can some of these be eliminated?

**Limitations:**

The authors need to be clear about whether their results actually (i) imply that CRN is useful for RL and (ii) the reason is gradient dominance. If they could actually empirically verify the reason is gradient dominance this would be a nice paper.

**Strengths And Weaknesses:**

The title of the paper is stochastic second-order methods provable beat SGD for gradient-dominated functions. However, the claim that SCRN has a better worst-case complexity that SGD on gradient dominated is not proven --- that would require proving a lower bound on the convergence on the SGD for gradient dominated functions which is not provided.

Results showing that SCRN performs well on RL problems are very interesting. However, it is implied by the authors that the explanation is that it is driven by gradient dominance. The authors need to make this hypothesis explicit otherwise the connection between the experiments and theory is weak. I personally am skeptical that gradient dominance explains the performance on RL problems. This is a comparison on worst-case bounds which may not explain actual performance. The authors do not run any experiments to test this hypothesis that gradient dominance is driving performance, e.g., estimating the magnitude of gradient dominance of various functions (i.e., estimating the alpha parameter) and showing SCRN does better compared with SGD for alpha = 1/2 versus alpha = 2.

Next, the results are not as surprising as the title and abstract in my view make it appear. In particular, for SCRN already has a better known worst-case complexity of O(\eps^{-3.5}) [3] compared with O(\eps^{-4}) for SGD when gradient domination does not hold. Moreover, gradient domination combined with well-known generic proof techniques [15,25] can be used to improve such complexity bounds.

**Writing issues:**

The paper also should emphasize that SCRN requires the Hessian to be Lipschitz whereas the SGD bound does not. This should be apparent in Table 1.

Theorem statements are poorly written. For example, Theorem 2 & 3 are one long sentence, please split into multiple sentences and make them more readable. Theorem 3 is particularly bad:
1. "Under gradient dominance property with \alpha = 1, Assumptions 1, 3 (for alpha = 1) and 5, variance reduced SCRN ..." =>
"Assume that Assumption 1, 3 (with alpha=1) and 5 hold. Also assume the gradient dominance property holds with alpha = 1. Then variance reduced SCRN ... "
2. expectation and average are synonyms, so you are effectively writing "on average by making ... on average, respectively". Please also try to be more precise so there is no ambiguity in the theorem statements.

**Update after author feedback**

Thanks for the author response. I have slightly raised my score. I still think many of my points about theory still stand and the writing needs a lot of improvement, but I did appreciate the authors alleviating my concerns about the experiments.

---

> ### Author Response · Authors · 2022-08-02
> **Response to Reviewer UAyg (Part 1)**
>
> > The title of the paper is stochastic second-order methods provable beat SGD for gradient-dominated functions. However, the claim that SCRN has a better worst-case complexity that SGD on gradient dominated is not proven --- that would require proving a lower bound on the convergence on the SGD for gradient dominated functions which is not provided.
>
> Thanks for the comment. We agree with the reviewer that the title was imprecise. In
> the abstract and the main body of the paper, we explicitly mentioned that SCRN improves
> upon the best-known sample complexity of SGD. To emphasize this in the title too, we have changed it to “Stochastic Second-Order Methods Improve Best-Known Sample Complexity of SGD for Gradient-Dominated Functions”. It is noteworthy that deriving a lower bound on the sample complexity under gradient dominance property (even in the deterministic case) is still an open problem.
>
> > Results showing that SCRN performs well on RL problems are very interesting. However, it is implied by the authors that the explanation is that it is driven by gradient dominance. The authors need to make this hypothesis explicit otherwise the connection between the experiments and theory is weak. I personally am skeptical that gradient dominance explains the performance on RL problems. This is a comparison on worst-case bounds which may not explain actual performance.
>
> The improvement of sample complexity of SCRN upon the best-known sample complexity of SPG (see Remark 8) motivated us to perform experiments in various RL settings. However, in the experiments, we did not assert that the significant performance of SCRN in the grid environment is mainly due to the gradient dominance property. In fact, we investigated why SCRN quickly finds a path to the goal. As we mentioned in lines 310-312 of the submitted version, SPG and REINFORCE get stuck in the initial state for some period of time, while SCRN quickly escapes the flat plateau. We also provided some demonstrations in the supplementary file to show this phenomenon. One possible explanation is that SCRN uses the curvature information in the Hessian matrix to escape the flat plateau. Similar behavior has been observed before in the literature for natural policy gradient [1].
>
> [1] Kakade, Sham M. "A natural policy gradient." Advances in neural information processing systems 14 (2001).
>
> > The authors do not run any experiments to test this hypothesis that gradient dominance is driving performance, e.g., estimating the magnitude of gradient dominance of various functions (i.e., estimating the alpha parameter) and showing SCRN does better compared with SGD for alpha = 1/2 versus alpha = 2.
>
> In the submitted version, we did experiments in Appendix A.2.3 on some synthetic functions that satisfy gradient dominance with $1<\alpha\le2$ and showed to what extent SCRN improves upon SGD for various $\alpha$. Besides, we evaluated the impact of $\tau_F$ on the performances of SCRN and SGD which further validates our analysis.

---

> ### Author Response · Authors · 2022-08-02
> **Response to Reviewer UAyg (Part 2)**
>
> > Next, the results are not as surprising as the title and abstract in my view make it appear. In particular, for SCRN already has a better known worst-case complexity of $O(\epsilon^{-3.5})$ [3] compared with $O(\epsilon^{-4})$ for SGD when gradient domination does not hold. Moreover, gradient domination combined with well-known generic proof techniques [15,25] can be used to improve such complexity bounds.
>
> Regarding the results: The improvement of $\mathcal{O}(\epsilon^{-0.5})$ in the general non-convex setting does
> not imply that the same holds for a specific class of functions (here, the gradient-dominated
> functions). In fact, for $\alpha=2$, there is no improvement upon SGD and the results in
> this paper are in part interesting because they show for which values of $\alpha$, SCRN improves the best-known sample complexity of SGD. Surprisingly, the largest improvement is for $\alpha=1$, which is incidentally the value for which a weak version of SCRN holds in some RL applications (see Section 4). Moreover, we proposed a variance-reduced method to further improve the sample complexity for the case of $\alpha=1$ to $\mathcal{O}(\epsilon^{-2})$. We also studied the impact of inexact sub-solver and showed that it is just needed to converge to an approximate first-order stationary point (Remark 6), while the general non-convex setting requires to approximate the exact solution of the sub-problem [33].
>
> Regarding the proof techniques: 1- We provided a stochastic analysis, in which separating the estimation error terms in the additive form in the recursion is a key ingredient to carry out our analysis (Lemma 1).  It is unclear how the proof techniques in [15,25] can be applied to study our problem:  as [25] deals with the deterministic case and [15] with first-order methods. 2- In order to achieve $\epsilon$-global stationary point, one way is to use fixed batch-sizes such that the error terms are bounded in the order of $\epsilon$ in each iteration. However, we showed that it suffices to make sure that the error terms are decaying at the same rate as the function values (Lemma 10 in Appendix A.3). Combining this idea with a variance-reduction technique yields a novel method that reduces the sample complexity to $\mathcal{O}(\epsilon^{-2})$ for $\alpha=1$. 3- We proposed an expectation analysis for SCRN under gradient dominance property with $1\leq \alpha \leq 3/2$, which does not need the bounded centered gradient and Hessian estimators which are assumed before for analyzing SCRN in general non-convex setting. Under gradient dominance condition with $1\leq \alpha \leq 3/2$, dependency of Hessian sample complexity on dimension is mitigated from $poly(d)$ to $\log d$ ($d$ is the dimension of the ambient space) by using Lemma 3 in Appendix.
>
> > Writing issues: The paper also should emphasize that SCRN requires the Hessian to be Lipschitz whereas the SGD bound does not. This should be apparent in Table 1.
>
> Thanks for the suggestion. We mentioned the required assumptions for each method in the caption of Table 1.
>
> > Theorem statements are poorly written. For example, Theorem 2 and 3 are one long sentence, please split into multiple sentences and make them more readable. Theorem 3 is particularly bad: 1. "Under gradient dominance property with $\alpha = 1$, Assumptions 1, 3 (for alpha = 1) and 5, variance reduced SCRN ..." to "Assume that Assumption 1, 3 (with alpha=1) and 5 hold. Also assume the gradient dominance property holds with alpha = 1. Then variance reduced SCRN ... " 2. expectation and average are synonyms, so you are effectively writing "on average by making ... on average, respectively". Please also try to be more precise so there is no ambiguity in the theorem statements.
>
> We revised the statements of Theorem 2 and 3 as suggested by the reviewer.
>
> > Questions: Is it not already known that SCRN performs empirically well on RL problems? I am not an expert on optimization methods for RL but it seems surprising. Could you perhaps suggest why no other papers have tested SCRN on RL problems.
>
> To the best of our knowledge, there is no previous work on applying SCRN in RL settings. This might be explained by the debate, running even in the community of optimization, on whether stochastic second-order methods have advantages over stochastic first-order methods. Here, to the best of our knowledge for the first time we established that SCRN under gradient dominance property, improves the sample complexity over SGD for $\alpha\in [1, 2)$. The largest improvement is for the case of $\alpha=1$, which is the case that applies in RL.
>
> > Does your implementation work poorly on non-RL problems?
>
> SCRN works fairly well in supervised learning tasks and its performance in training deep neural networks has been reported in [33]. In the experiments made in our paper, our focus is mainly on various RL settings when the gradient dominance property is satisfied.

---

> > ### Comment · Reviewer_UAyg · 2022-08-08
> > **Practicality of cubic regularized Newton in ML**
> >
> > I am still a little hung up on why your results are so good for RL. They almost appear too good to be true. I mean if you are this much better than state-of-the-art why shouldn't everyone use SCRN? This is in contrast, in my experience, to the poor performance of SCRN on supervised learning problems.
> >
> > > SCRN works fairly well in supervised learning tasks and its performance in training deep neural networks has been reported in [33]. In the experiments made in our paper, our focus is mainly on various RL settings when the gradient dominance property is satisfied.
> >
> > I disagree that SCRN works well on supervised deep learning learning tasks. [33] does not test on supervised learning problems (the MNIST experiment is an autoencoder). Indeed, SCRN is not used at all in practice for supervised learning despite almost half a decade of intensive research (please reference a paper with strong empirics that indicates otherwise --- I would be very interested). Moreover, there are good reason to believe that second-order methods are not natural methods for deep learning, e.g., see https://iclr.cc/virtual/2021/poster/2577.
> >
> > I understand that perhaps RL is different ... I am just very surprised no one else has tried this. I didn't find your explanation in the rebuttal very convincing.
> >
> > Also for the example 1 and 2 in the appendix. Are they stochastic?

---

> > > ### Author Response · Authors · 2022-08-08
> > > **Cubic Regularized Newton in RL**
> > >
> > > We thank the reviewer for reading the rebuttal. We should emphasize the following remarks regarding the differences between the RL tasks considered in this paper and deep learning (DL):
> > > * Structured nonconvexity and landscape - unlike DL, the nonconvex objective of RL satisfies the non-uniform gradient dominance property and is smooth under softmax parametrization, whereas the nonconvex objective of DL could suffer from (exponentially many) spurious saddle points and is often nonsmooth when practical ReLU activations are used.
> > > * Local geometry matters as we are learning the optimal policy, which lies on distribution spaces. Exploiting local geometry and curvature information greatly improves the convergence and exploration, as evidenced by practical state-of-the-art algorithms such as natural policy gradient (which can be viewed as a Newton-like method) and its variants such as TRPO (see Section 6 in [1])
> > > * Mild overparameterization (network complexity) is required for tabular RL tasks and continuous control tasks with moderate dimensions; thus, computing Hessian vector multiplication is much more affordable.
> > > * There are also some recent works [2,3] using second-order information in variance-reduced policy gradient methods, showing remarkable performance in continuous control tasks.
> > >
> > > Regarding experiments of SCRN or other its variants on supervised learning tasks, there exist some other works such as [4,5] where they performed extensive experiments for the task of logistic regression on several real datasets.
> > >
> > > Moreover, Examples 1 and 2 in Appendix are in the stochastic setting.
> > >
> > > [1] Schulman, John, et al. "Trust region policy optimization." International conference on machine learning. PMLR, 2015.
> > >
> > > [2] Shen, Zebang, et al. "Hessian aided policy gradient." International conference on machine learning. PMLR, 2019.
> > >
> > > [3] Salehkaleybar, Saber, et al. "Adaptive Momentum-Based Policy Gradient with Second-Order Information." arXiv preprint arXiv:2205.08253, 2022.
> > >
> > > [4] Kohler, Jonas Moritz, and Aurelien Lucchi. "Sub-sampled cubic regularization for non-convex optimization." International Conference on Machine Learning. PMLR, 2017.
> > >
> > > [5] Zhou, Dongruo, Pan Xu, and Quanquan Gu. "Stochastic Variance-Reduced Cubic Regularization Methods." J. Mach. Learn. Res. 20.134 (2019): 1-47.

---

> > > > ### Comment · Reviewer_UAyg · 2022-08-09
> > > > **Cubic Regularized Newton in RL**
> > > >
> > > > Thanks for the response!

---

> ### Author Response · Authors · 2022-08-02
> **Response to Reviewer UAyg (Part 3)**
>
> > How are the proof techniques different from standard techniques [15,25]?
>
> Please refer to the answer in Part 2.
>
> > Have you experimented with variance reduced SCRN? How does it compare with SCRN?
>
> We performed experiments for variance-reduced SCRN in control tasks and added the results in Appendix A.5. In Humanoid, Reacher, and Walker environments, the variance-reduced SCRN achieves the average episode return of about $450$, $-10$, and $400$ with lower system probes compared with SCRN, respectively.
>
> > Are all the problems you test on twice-differentiable? Is that typical in RL?
>
> In the RL setting, the value functions induced by common policies such as softmax tabular and Gaussian policies are at least twice differentiable with respect to the parameter $\theta$.
>
> > The algorithm presented by the authors requires three highly problem-dependent hyperparameters. Can some of these be eliminated?
>
> Regarding the cubic penalty term, one can utilize adaptive-line search strategies similar to  [25]. Regarding the batch-sizes, as we mentioned in Section 3.3, for $1\leq \alpha< 3/2$, it suffices to make sure that the error terms at iteration $t$ are  $\mathcal{O}(t^{-(2\alpha)/(3-2\alpha)})$, which equals the convergence rate of the function values $F(\mathbf{x}_{t})-F(\mathbf{x}^*)$. Thus, in practice, we can use this rate to adjust the batch-sizes along the iterates.
>
> > Limitations: The authors need to be clear about whether their results actually (i) imply that CRN is useful for RL and (ii) the reason is gradient dominance. If they could actually empirically verify the reason is gradient dominance this would be a nice paper.
>
> We should emphasize that the main goal of this paper is to study the performance of SCRN under gradient dominance property for $\alpha\in [1, 2]$. We showed that SCRN improves the sample complexity of SGD for $1\leq \alpha <2$. We applied SCRN in the RL setting because sample efficiency is more important than the computational complexity in this application, which also justifies why we are using a second-order method. Indeed, SRCN is useful in RL in settings such as grid environments, where it significantly outperforms first-order methods by exploiting second-order information to escape flat plateaus.

---

> ### Author Response · Authors · 2022-08-08
> **Author-Reviewer discussion period**
>
> Dear Review UAyg, as the discussion period is ending soon, we would like to know if we have addressed your concerns about the limitations.  Do you have any further questions that we can clarify?

---

### Official Review · Reviewer_riBm · 2022-07-11

**Rating:** 6
**Confidence:** 4
**Soundness:** 3 good
**Presentation:** 3 good
**Contribution:** 3 good

**Summary:**

The paper provides sample-complexity bounds for stochastic second order methods that improve the best known bounds for first order methods. In particular, the paper considers a stochastic version of Nesterov & Polyak Cubic-regularized Newton’s method. The paper studies the sample complexity of the algorithm for the class of gradient dominated function. The sample complexity is shown to improve the best known bounds for first-order methods. The problem class covers many policy gradient methods of interest, which are commonly used in reinforcement learning. The experiments show good performance when applied to RL problems.

**Questions:**

It is nice that it is possible to improve sample-efficiency, but this seems to come at the cost of computational and memory complexity. Some discussion on these trade-offs would strengthen the paper.

Regarding the sample complexity for policy gradient in Section 4. How do these bound compare to the state-of-the-art for policy gradient? Discussion on this would be beneficial.

Please specify the notation x^*. Is it local or global optimal solution? Does gradient dominance imply that all local optimal solutions are global optimal? or that there is a unique optimal solution?

**Limitations:**

No major limitations as far as I can tell.

**Strengths And Weaknesses:**

Strengths: The paper improves the state-of-the-art sample complexity by using second order information. The result are quite interesting given the high interest in ML in capitalizing on second-order information in stochastic optimization, which is usually difficult to achieve. The experimental results are quite impressive, the approach largely improves the performance of vanilla policy gradient, some times of order of magnitude. Overall, the theoretical results seem quite solid.

Weaknesses: I feel that title of the paper is bit of an over statement. The results show that the paper achieves better sample complexity for SCRN than has been achieved with SGD. However, these bounds could be loose. The paper does not discuss the tightness of these bounds.

There are a lot of repetitions in the introduction and related works. B) Equation 2 is repeated again in 5. C) In line 134, the phrase "defined in (35)" is ambiguous, which seems to clear up the ambiguity by referring to the appendix also. Also, I think that in line 162, the dimensions of the matrix is d by d, and as a result, the matrix 0 must have dimensions d-1 by d-1 not d by d

There are some limitations with the experiments. The lack of the comparison with the state-of-the-art methods (unlike MBPG and HAPG, which are claimed to be the state-of-the-art algorithms, while they are unknown among the RL community,) that have been developed based on the principles of MM optimization, like SAC, PPO. Such a these algorithms, without the need to have strong assumptions, have been able to achieve good results in terms of sample complexity by keeping the computational complexity at an acceptable level.  For the maze environments it would be more meaningful to plot SPG with entropy regularization and REINFORCE with entropy reqularization results. There is also an inadequate discussion of results on maze environments and robotic control tasks.

---

> ### Author Response · Authors · 2022-08-02
> **Response to Reviewer riBm (Part 1)**
>
> > Weaknesses: I feel that title of the paper is bit of an over statement. The results show that the paper achieves better sample complexity for SCRN than has been achieved with SGD. However, these bounds could be loose. The paper does not discuss the tightness of these bounds.
>
> Thanks for pointing it out and we fully agree with the comment. In the abstract and the main body of the paper, we explicitly mentioned that SCRN improves upon the best-known sample complexity of SGD. To reflect this in the title, we decided to change the title to "Stochastic Second-Order Methods Improve Best-Known Sample Complexity of SGD for Gradient-Dominated Functions". It is noteworthy that deriving a lower bound on the sample complexity under gradient dominance property (even in the deterministic case) is still an open problem.
>
> > There are a lot of repetitions in the introduction and related works. B) Equation 2 is repeated again in 5. C) In line 134, the phrase "defined in (35)" is ambiguous, which seems to clear up the ambiguity by referring to the appendix also. Also, I think that in line 162, the dimensions of the matrix is d by d, and as a result, the matrix 0 must have dimensions d-1 by d-1 not d by d.
>
> We kept only equation (5) and removed any repetitions. The constants in Lemma 1 are too lengthy to be placed in the main text, and we prefer instead to place a pointer to equation (35) which presents their explicit expressions in the appendix. Regarding the dimension of matrix in line 162, please note that the dimension of the gradient vector is $d$, hence the dimension of this matrix is $(d+1)\times (d+1)$.
>
> > There are some limitations with the experiments. The lack of the comparison with the state-of-the-art methods (unlike MBPG and HAPG, which are claimed to be the state-of-the-art algorithms, while they are unknown among the RL community,) that have been developed based on the principles of MM optimization, like SAC, PPO. Such a these algorithms, without the need to have strong assumptions, have been able to achieve good results in terms of sample complexity by keeping the computational complexity at an acceptable level.
>
> The main goal of this paper is to establish the theoretical performance of SCRN under gradient
> dominance property and compare it mainly to SGD. As such our numerical experiments are designed to showcase the theoretical results and illustrate the benefit of using stochastic second-order information. To be clear, we do not intend to beat the state-of-the-art algorithms for RL. Note that the state-of-art RL algorithms like SAC and PPO consider different objectives such as clipped objective in PPO and maximum entropy objective in SAC, and rely on other algorithmic advances in their implementation, with limited theoretical guarantees. Hence, it is hard to make a fair comparison with them. Instead, our experiments mainly focus on comparison with stochastic first-order methods designed for the same objective: in grid environments, we compared SCRN with SPG and REINFORCE, and in the control tasks, we compared SCRN with two recent variance-reduced variants of SGD. We hope that our paper could serve as a starting point to investigate better-tuned second-order approaches to learning problems at large and RL in particular.
>
> > For the maze environments it would be more meaningful to plot SPG with entropy regularization and REINFORCE with entropy regularization results.
>
> We evaluated these two algorithms with entropy regularization and the improvements were not significant enough compared with the case without entropy regularization in the maze environments. Nevertheless, we added Figure 6 in Appendix A.5 in the revised version and discussed the results.
>
> > There is also an inadequate discussion of results on maze environments and robotic control tasks.
>
> Due to limited space, we added more discussion on the experimental results in Appendix A.5.
>
> > Questions: It is nice that it is possible to improve sample-efficiency, but this seems to come at the cost of computational and memory complexity. Some discussion on these trade-offs would strengthen the paper.
>
> In the submitted version, we provided the computational complexity of SCRN in Remark 6 (lines 201-206). Using Pearlmutter’s algorithm, we can compute the Hessian vector product in $\mathcal{O}(d)$ both in time and memory. The total computational complexity of SCRN is $\mathcal{O}(dT\epsilon^{-3/\alpha})$, where $T$ is the number of iterations of SCRN, while the complexity of SGD is $\mathcal{O}(d\epsilon^{-4/\alpha+1})$. Therefore, the computational complexity of SGD is less than the one of SCRN by a factor in the order of $\epsilon^{-1/(2\alpha)}$ for $\alpha\in[1,3/2)$, and by a factor in the order of $\epsilon^{-1+1/{\alpha}}$ for $\alpha\in[3/2,2]$.
> We added this discussion to Remark 6 (lines 205-209).

---

> ### Author Response · Authors · 2022-08-02
> **Response to Reviewer riBm (Part 2)**
>
> > Regarding the sample complexity for policy gradient in Section 4. How do these bound compare to the state-of-the-art for policy gradient? Discussion on this would be beneficial.
>
> As we mentioned in Remark 8, under weak gradient dominance with $\alpha=1$ (Assumption 6), the best known sample complexity of SPG is  $\tilde{\mathcal{O}}(\epsilon^{-3})$ when $\epsilon'=0$ and $\tilde{\mathcal{O}}(\epsilon^{-1}\epsilon'^{-2})$ when $\epsilon'>0$ [40, Theorem C.1]. Therefore, SCRN improves upon the best-known sample complexity of  SPG in both cases $\epsilon'=0$ and $\epsilon'>0$ by a factor of $\mathcal{O}(\epsilon^{-1/2})$. We also discussed the sample complexity of CRN and its comparison with the best-known sample complexity of policy gradient in Remark 9.
>
> > Please specify the notation $\textbf{x}^*$. Is it local or global optimal solution? Does gradient dominance imply that all local optimal solutions are global optimal? or that there is a unique optimal solution?
>
> Here $\textbf{x}^*$ is a global optimal solution, and because of the gradient dominance property, all local minima are globally optimal. The global solutions are not necessarily unique. We clarified this in the revised version (line 123).

---

> ### Author Response · Authors · 2022-08-08
> **Author-Reviewer discussion period**
>
> Dear Review riBM, as the discussion period is ending soon, we would like to know if we have addressed your comments regarding the sample-efficiency.  Do you have any further questions that we can clarify?

---

### Meta-Review · Area_Chair_9Pgk · 2022-08-26

**Recommendation:** Accept
**Confidence:** Less certain

**Metareview:**

This paper presents improved convergence rates for SCRN on gradient dominated functions. Reviews all agree that the paper advances known results. Please take the comments about outperforming SGD into account in the final copy.

**Award:**

No

---

### Decision · Program_Chairs · 2022-09-14

Accept